# The Influence of Various Welding Methods on the Microstructure and Mechanical Properties of 316Ti Steel

**DOI:** 10.3390/ma17071681

**Published:** 2024-04-06

**Authors:** Piotr Noga, Tomasz Skrzekut, Maciej Wędrychowicz, Marek St. Węglowski, Marcel Wiewióra

**Affiliations:** 1Faculty of Non-Ferrous Metals, AGH University of Krakow, A. Mickiewicza Av. 30, 30-059 Krakow, Poland; pionoga@agh.edu.pl (P.N.); skrzekut@agh.edu.pl (T.S.); wiewiora@agh.edu.pl (M.W.); 2Faculty of Mechanical Engineering, Institute of Materials and Biomedical Engineering, University of Zielona Gora, Prof. Z. Szafrana Street 4, 65-516 Zielona Góra, Poland; 3Łukasiewicz—Upper Silesian Institute of Technology, Karola Miarki Str. 12-14, 44-100 Gliwice, Poland; marek.weglowski@git.lukasiewicz.gov.pl

**Keywords:** 316Ti steel, MIG, TIG, EBW and PAW welding, mechanical properties

## Abstract

Austenitic stainless steels are very popular due to their high strength properties, ductility, excellent corrosion resistance and work hardening. This paper presents the test results for joining AISI 316Ti austenitic steel. The technologies used for joining were the most popular welding techniques such as TIG (welding with a non-consumable electrode in the shield of inert gases), MIG (welding with a consumable electrode in the shield of inert gases) as well as high-energy EBW welding (Electron Beam Welding) and plasma PAW (plasma welding). Microstructural examinations in the face, center and root areas of the weld revealed different contents of delta ferrite with skeletal or lathy ferrite morphology. Additionally, the presence of columnar grains at the fusion line and equiaxed grains in the center of the welds was found. Microstructural, X-ray and ferroscope tests showed the presence of different delta ferrite contents depending on the technology used. The highest content of delta ferrite was found in the TIG and PAW connectors, approximately 5%, and the lowest in the EBW connector, approximately 2%. Based on the tests carried out on the mechanical properties, it was found that the highest properties were achieved by the MIG joint (R_m_, 616, R_p0.2_ = 335 MPa), while the lowest were achieved by the PAW joint (R_m_ = 576, R_p0.2_ = 315 MPa).

## 1. Introduction

Austenitic chrome-nickel steels belong to the group of corrosion-resistant metal materials. They are distinguished by non-magnetic properties, high susceptibility to plastic deformation and good technological features. Due to the fact that they are single-phase materials, they cannot be strengthened as a result of heat treatment, but only by cold deformation. Metallurgical products made from these steels are used in installations of the petrochemical and chemical industries, in the pulp and paper industry, in the food industry and in oil and gas extraction. Austenitic steels may contain up to 10% ferrite. The main components of these steels are Fe, Cr (17–25%), and Ni (8–30%), and include elements such as C, N, Mn, Mo and Si. Austenitic steels belong to a large group of stainless steels [1,2].

The structure of corrosion-resistant steels can be obtained by optimal selection of alloying elements that favor the formation of both austenite and ferrite. The high price of nickel makes Cr-Ni austenitic steels expensive. The relatively low strength properties of the γ phase force a constant search for solutions that could partially replace nickel with other elements and, at the same time, improve the corrosion resistance and mechanical properties of steel. As a result of the introduction of alloy additions to the steel, the stability ranges of the γ phase change significantly. In these steels, the structure is shaped by the amount of austenite-forming elements such as Ni, N, C, Mn, and Cu, which expand the concentration range stability of the γ phase, as a result of which the amount of nickel can be correspondingly lower [3].

In order to improve mechanical properties and corrosion resistance, ferrite-forming elements such as Mo, W, Si, Ti, Nb and V are used as alloying additives. When introduced into steel, they also narrow the stability range of the γ phase, usually limiting the tendency of austenite to release phases: σ, Laves, χ and others. Maintaining the single-phase γ structure, therefore, requires introducing an optimally high Ni content into the steel. Otherwise, the steel may obtain a two-phase structure: austenite and metastable ferrite. Literature data confirm that a small amount of δ ferrite prevents hot cracking in austenitic steels, and an amount in the range of 1–6% prevents hot cracking in welded austenitic steels and reduces the crack growth rate during stress corrosion cracking [3,4,5,6,7,8,9].

The amount of ferrite in the weld is determined based on its chemical composition. Alloying elements in austenitic steels are divided into ferrite-forming and austenite-forming elements. The influence of the chemical composition on the microstructure and thus the solidification method is expressed in chromium and nickel equivalents. The chromium equivalent is equal to the sum of the Cr content and ferrite-forming components multiplied by coefficients symbolizing the influence of these components on the relative amount of ferrite in relation to the influence of chromium. Figure 1 presents a Schaeffler diagram, showing the areas where the weld structure of corrosion-resistant steels occurs, expressed in terms of nickel and chromium equivalents [10,11]. This diagram is a helpful tool for determining the effect of mixing the filler metal with the base material of the joint, allowing the calculation of the share of ferrite in the joints.

As can be seen from Figure 1, the Schaeffler graph does not take into account nitrogen as an element stabilizing austenite, but carbon is assigned a 30-fold stronger influence than nickel. However, the Schaeffler graph should not be considered indisputable because, for example, the amount of ferrite decreases as the cooling rate of the junction decreases. The graph allows calculation of the tendency to create ferrite, but does not take into account the technological parameters or operating conditions of the structure welded. In some cases, it is required that the amount of ferrite in the connector remains within certain limits [12]. In 1973, Long and Delong developed a chart in which they not only took into account the influence of nitrogen on the weld structure but also changed the inclination of the line with constant ferrite content, which makes their chart different from the Scheaffler diagram [13].

The relative volume of ferrite in the structure of an austenitic weld metal is referred to as the ferrite number (FN). This value is determined on the basis of magnetic measurements, which is enabled by the ferromagnetic features of delta ferrite with a with a body-centered cubic (BCC). However, austenite with a face-centered cubic (FCC) is paramagnetic. This method is a comparative method. The basis of this measurement is the value of the force needed to detach a specific permanent magnet with known coercivity from an austenitic deposited metal. FN values correlate with the amount of ferrite expressed as a percentage only up to a value of approximately 7. Ferritometers are scaled on standards in the range of 3–28 FN, which is sufficient to evaluate austenitic weld metals. For higher ferrite contents, an extended ferrite number EFN was introduced, which is a continuation of the FN scale [14].

Model austenitic corrosion-resistant steels are characterized by good weldability, and are characterized by significant strength, corrosion resistance, good impact strength and ductility in the weld area. When making welded joints, remember the following properties of austenitic steels:The coefficient of thermal expansion compared to steel with a ferritic structure is approximately 50% higher, which favors the formation of residual stresses and deformations.The thermal conductivity in these steels is approximately 60% lower, as a result of which the heat is concentrated in the welding zone. The way to dissipate it is to use copper washers [15,16,17,18].

Austenitic stainless steels have a tendency to hot crack during processing crystallization. This negative phenomenon, which worsens the mechanical properties of the weld, is counteracted by appropriate modifications of the chemical composition of the weld metal, which ensure obtaining up to approximately 10% of ferrite in the weld. As a result of heating in the temperature range of 550–900 °C, brittleness may occur due to the decomposition of ferrite to form a brittle σ phase. The lack of resistance to intergranular corrosion of joints welded at elevated temperatures is improved by stabilization with titanium or niobium, lowering the carbon content or post-weld heat treatment [19].

Caution should be exercised when welding austenitic steels with a high content of alloying elements. This applies especially to steels with a high content of Cr and Mo, because these elements tend to segregate in the weld, which results in the formation of zones depleted in these elements. For this reason, binders with higher chromium and molybdium content are used, taking into account segregation to ensure a minimum content of alloying elements in the weld. However, the increased content of Cr and Mo additionally intensifies the precipitation processes of brittle phases (Table 1), which reduces corrosion resistance when held at high temperatures [20,21].

These negative processes are prevented by increasing the speed cooling that reduces the amount of heat introduced during welding. It is also important that a higher content of chromium and nickel reduces the solubility of carbon in the alloy. If the welds produced in the joints are fully austenitic, hot cracking may occur. The tendency to cracking can be counteracted by using binders with low impurity content, lowering the temperature and limiting the amount of heat introduced [22].

Although numerous publications can be found on welding of type 316 corrosion-resistant steels, in the case of 316Ti, steel there is no comprehensive comparison of the joints in terms of microstructural and mechanical properties depending on the method used. In [23], the authors performed a comparative analysis of welded joints made of AISI 316L, AISI 316Ti, AISI 304L and AISI 321 steel obtained by microplasma welding. The analysis showed that for the same welding conditions, the highest strength properties were achieved for welded joints made of 304L and 316Ti steel. The 316L steel joint was characterized by the lowest strength properties, while the 321 steel joint had the lowest hardness in the welded joint area. Taban et al. [24] using the plasma welding method, they welded sheets of corrosion-resistant steel 316Ti. Welding were made without filler material and using filler material (316L steel). Tensile, bending and impact tests were carried out at temperatures from −60 °C to 20 °C. Additionally, the chemical composition in the weld area was analyzed and the share of delta ferrite was measured. The measured delta ferrite content was in the range of 9–13%, which is the permissible value according to international standards. The authors found that corrosion-resistant steel 316Ti can be welded with or without filler material. The obtained mechanical properties were at a comparable level. In this work, the joint areas: face, center and root of the weld were examined in detail. The delta ferrite content was calculated and compared depending on the joint area as well as depending on the research method used. Additionally, these values were compared with theoretical values read from the Schaeffer diagram.

## 2. Materials and Methods

The material used in the tests were plates with dimensions: 6.35 × 1000 × 2000 mm made of austenitic steel AISI 316Ti, 1.571, X2CrNiMoTi17-12-2, acc. PN-EN 10088-2:2014-12 (producer OUTO KUMPU, Helsinki, Finland). The chemical composition of the steel based on the acceptance certificate is presented in Table 2. The additional material used in MIG and TIG welding was AISI 316Si steel. The chemical composition of the additional material is given in Table 3.

Sheets measuring 150 × 300 mm were prepared to make test joints. The welding process was performed along the edge parallel to the direction of rolling of the sheets. The welding processes were performed at the Welding Institute (now Lukasiewicz—Upper Silesian Institute of Technology, the Welding Centre) in Gliwice. The following steel joining processes were performed:Metal Inert Gas—MIG;Tungsten Inert Gas—TIG;Electron Beam Welding—EBW;Plasma Arc Welding—PAW.

The final welding parameters were applied based on preliminary technological tests on the specimens under the dimensions. The main goal of the technological trials was to achieve the welded joints without welding imperfections according to standards (e.g., EN 5817). After tests, the final welded joints (300 × 300 mm) were welded and tested. The selected final welding parameters are given in Table 4. Additionally, for MIG, TIG as well as PA welding, 100% Ar as shielding gas was applied. Moreover, the filler material AISI 316Si was also used. The EBW process, without shielding gas (in vacuum 2 × 10^−5^ bar) and filler material, was carried out.

The microstructure of the welded joints was examined using an Olympus GX51 light microscope. The samples were cold mounted in a Struers FixiForm container using Struers’ EpoFix epoxy resin. The samples prepared in this way were then ground on sandpaper with a grit of 320–1500. Diamond suspensions were used for polishing: DP-Suspension P 9 μm, 3 μm, 1 μm, and the microsections were made on a RotoPol-11 grinder and polisher with a RotoForce-1 Struers head. Finishing polishing was carried out using OP-S polishing suspension from Struers. In order to reveal the microstructure of AISI 316Ti steel, the specimens were electrolytically etched in 10% nitric acid at a voltage of 1.5 V for 20 s at room temperature. Scientific research on a micro scale was performed on a HITACHI SU-70 scanning electron microscope with an attachment for X-ray microanalysis of chemical composition (EDS). The chemical composition was determined in the micro-areas of the weld face, weld center and weld root, as well as in the heat-affected zone. Tests for measuring the ferrite content in the base material, in the welds of ALSI 316Ti steel, were carried out using the quantitative metallography method and using a MPD-100 ferritoscope. The ferritoscope uses a method based on the principle of measuring the magnetic properties of a material. The amount of ferrite is calculated from the magnetic permeability of the tested material. The results are given in percentages.

Mechanical tests of the obtained joints included a uniaxial tensile test, bending test and hardness measurement using the Vickers method. The width, thickness and length of the measurement base in the case of uniaxial tensile samples were: 25 mm × 6.35 mm × 60 mm. Tensile tests were performed on three samples taken from each welded joint. The tests were performed on the MTS Criterion C45 testing machine according to PN-EN ISO 4136:2022-12 [25]. Flat samples with dimensions of 300 × 25 × 6.35 mm were taken for bending tests. The tests were performed in accordance with the recommendations of PN-EN ISO 5173:2010 [26]. A Shimadzu HMV-G hardness tester was used to test the hardness. Measurements were performed when the indenter was loaded with a force of 19.61 N (HV 2) for 10 s (PN-EN ISO 6507-2:2018-05) [27]. Measurements were made in four measurement lines that passed through the base material, the sample weld and the HAZ of each sample. Hardness measurements made in this way were used to prepare 3D maps in MATLAB and hardness profiles in welded joints. The dimensions of the samples used for testing are presented in Figure 2.

## 3. Results and Discussion

The microstructure of the base material of AISI 316Ti steel, presented in Figure 3, is characterized by a fine-grained austenitic structure with regular grains, the average grain diameter of which is 20 µm. Numerous narrow bands of ferrite arranged along the rolling direction of the sheet and annealing twins are visible, especially visible in the image obtained by Scanning Electron Microscopy (Figure 3A). The occurrence of delta ferrite as a result of the addition of titanium is confirmed by the element distribution map (Figure 3B) and the chemical composition analysis (Table 5). The addition of titanium in austenitic stainless steels is used to reduce the risk of intergranular corrosion during heating at temperatures of 425–815 °C. Titanium and carbon create carbides, which prevents the formation of chromium carbides. In this way, the chromium concentration remains unchanged and prevents the occurrence of intergranular corrosion. Otherwise, in the given temperature range, chromium carbides are formed, which are precipitated at the grain boundaries and in this area the concentration of this element drops to a dangerously low level. The created chromium carbides contribute to the occurrence of intergranular corrosion [28].

The microstructure of the MIG 316Ti welded joint is shown in Figure 4. In order to obtain the correct weld, two beads were made and marked with a dashed line. Additionally, the weld area was marked. Scanning microscopy showed that the crystallization front occurs from the fusion line towards the weld axis. It was also observed that the grain size depends on the distance from the fusion line; there are columnar grains right next to the fusion line, while in the weld axis the grains have an equiaxed shape. Figures obtained in the light microscopy mode, made in the area of face of weld, the weld center and the root of a weld, show the presence of an austenitic matrix, dark areas of delta ferrite are visible. Observations showed a narrow heat-affected zone, the average width of which was 115 µm, while the joint area covered an area of 51 mm^2^. No cracks or grain growth were noticed in the microstructure of the heat-affected zone. However, austenite grains were observed around which elongated ferrite grains formed a discontinuous mesh. At the fusion line, areas with an increased share of delta ferrite can be observed, which is characteristic during crystallization during welding of most austenitic steels. The amount of delta ferrite depends on the rate of solidification of the weld, which in turn depends on the amount of heat supplied during welding. The larger amount of delta ferrite at the fusion line is caused by the non-equilibrium cooling rate. In the microstructure of the weld, one can notice the presence of ferrite with lathy morphology as well as skeletal morphology. Skeleton ferrite is formed when the cooling rate is moderate, but as the cooling rate increases, the amount of delta ferrite increases and the framework ferrite is replaced by lathy ferrite. This is due to limited diffusion during the ferrite-austenite transformation [29]. In the areas of the face, center and root of the weld, a difference in structure can be noticed. Lathy ferrite can be observed near the fusion line and in the weld face area, which may indicate a higher cooling rate compared to other areas of the weld. The obtained microstructure is consistent with the results obtained in [30], in which the authors joined sheets of 316L austenitic steel using the MIG method. The authors found that the microstructure of the joint, the morphology of the delta ferrite and the width of the heat influence are determined by factors such as peak temperature, heating rate, time of maintaining the joint at elevated temperature and cooling rate. The element distribution map shown in Figure 4 confirmed the presence of delta ferrite rich in chromium and titanium carbides.

Figure 5 shows the microstructure of a TIG joint. In the case of the root bead and the center bead, the amount of energy introduced was similar, while the linear energy was lower by 0.39 kJ/min in the face bead. The difference in the amount of energy causes different grain sizes to be visible in individual layers of the weld, which decrease towards the face of weld. The observations showed a heat-affected zone, the average width of which was 174 µm, while the joint area covered an area of 42 mm^2^. The width of the heat-affected zone obtained in the welded joint using the TIG method is approximately 60 µm larger than the HAZ zone obtained using the MIG method. This is caused by the introduction of a larger amount of energy and is related to the different cooling speed. Microstructure researches using light microscopy showed the absence of cracks, bubbles, sticking, inclusions, proper penetration and correct formation of the face and root of the weld. Different amounts of heat introduced and different crystallization rates of the joint when applying subsequent layers are visible as narrow zones of enriched delta ferrite. In the areas of the fusion line, as in the case of MIG welding, delta ferrite can be observed, with an elongated morphology, while towards the weld axis the morphology of lamellar ferrite transforms into skeletal ferrite. The difference in morphology and distribution of delta ferrite occurs in the weld axis. In the root and center of the weld, the delta ferrite is “finer” and is evenly distributed compared to the delta ferrite present in the face of the weld. This is due to the uneven cooling speed of individual beads. In these areas, as in the case of MIG welding, different morphologies of delta ferrite can be distinguished: lathy and skeletal. Skeletal ferrite occurs both in the areas of the fusion line and in the face of weld, where the solidification rate is higher compared to other areas of the weld. In the root of the weld, the skeletal ferrite crystals are arranged in parallel, while in the areas of the center and face of the weld there is no clear orientation. The element distribution map made in the central part of the weld confirms the presence of delta ferrite rich in chromium and molybdenum and the occurrence of titanium carbides.

The microstructure of a PAW welded joint made of corrosion-resistant steel is illustrated in Figure 6. The joint is characterized by no welding imperfections along its entire length. The weld area was 35.5 mm^2^ and the average width of the heat-affected zone was 82 μm. Compared to the joints described above, this joint was the only one obtained in one bead, which involved the introduction of a large amount of heat during welding. The large amount of heat introduced during welding means that the interface between the base material and the weld is mild, and the occurrence of ferrite with skeletal morphology is negligible compared to TIG and MIG welded joints. In the area of the face of weld and the fusion line, the grains are finer than in the center and at the root of a weld. The qualitative analysis confirms the presence of austenite, delta ferrite enriched in Cr, Mo and titanium carbides. A similar microstructure of joints obtained by plasma welding was obtained by the authors in [24,31], in which they joined sheets of 7 mm and 10 mm thickness.

Figure 7 shows the microstructure of an electron beam welded joint made of 316Ti steel (EBW 316Ti). The parameters used allowed us to obtain a symmetrical weld with full penetration without welding imperfections. The analyzed structure is characterized by the smallest weld area of approximately 9 mm^2^ and the smallest width of the heat-affected zone (21 μm). In order to obtain the correct joint, a main bead and a cosmetic bead with circular oscillation were made. This is due to the lowest amount of heat input among the above-described joints made of 316Ti austenitic steel. The boundary between the base metal and the weld in the face, center and root of weld is sharp, and an increased amount of lathy ferrite can be noticed. Delta ferrite can be distinguished on the face of the weld, arranged in the direction of heat dissipation. At the border with the fusion line and at the border of the beads, where the temperature gradient is high, delta ferrite with a lathy morphology and elongated grains in the direction of heat dissipation can be noticed. In the weld axis, where the temperature gradient is smaller, the grains had an equiaxed shape. An element distribution map prepared similarly to the previous samples confirmed the presence of delta ferrite and titanium carbides. In [31], Kar et al. investigated the influence of electron beam oscillations on the mechanical properties and microstructure of 316Ti steel. The authors proved that the joints in which the beam oscillation was used showed a homogeneous microstructure compared to the joints without oscillation, which was due to better heat distribution. The difference in the linear energy of the beads and the width of the weld areas resulted in a larger amount of delta ferrite with lathy morphology in the face of weld compared to the root of weld.

The microstructures of the joints presented in this work are fully austenitic with delta ferrite precipitates. In the microstructures of the welds, cellular and dendritic microstructures can be observed, while in the weld axes there is an equiaxed structure. Columnar dendrites with a primary dendritic arm spacing (PDAS) and a secondary dendritic arm spacing (SDAS) were visible. A similar microstructure was observed in Kulkarni’s work [32]. The occurrence of dendritic microstructure at the fusion lines and equiaxed in the weld line can be explained by the lower ratio (G/R) of the temperature gradient (G) to the solidification rate (R) [33]. As a result, the center of the fusion zone exhibited a fine equiaxed dendritic structure.

Ferrite content measurements were made using the magnetic method using a ferritometer and the metallographic method. Based on the chemical composition of AISI 316Ti steel and AISI 316Si austenitic steel (filler metal), the chromium equivalent (Creq) and nickel equivalent (Nieq) were calculated. In the case of 316Ti steel, the nickel and chromium equivalents were 11.95 and 19.28, respectively, and for 316Si steel 12.50 and 22.15. After calculating these values, the predicted ferrite content in the weld was read from the Schaeffler graph (Figure 8) and it was read that the delta ferrite content was 8%. The value of the Creq/Nieq ratio has a significant impact on the solidification mode [34] and the resulting microstructure in the joint [35].

Figure 9 compares the theoretical content of delta ferrite, read from the Schaeffler chart, and the actual content obtained from ferritometer readings, as well as from metallographic calculations. Examples of weld microstructures after binarization are shown above bar charts with delta ferrite content. In the case of MIG and TIG joints, the theoretical delta ferrite content was 8%, while in the case of PAW and EBW welded joints, this value was equal to 5%. This difference is caused by the fact that in the case of the TIG and MIG methods, a filler metal with a higher content of alloy additives was used, which resulted in a higher chromium and nickel ratio. In the case of EBW and PAW joints, no filler metal was used. In relation to the delta ferrite content, calculated using the metallographic method, a decreasing trend of delta ferrite content from the face of weld towards root of weld. The ferrite content values obtained by the magnetic method are slightly lower than those obtained by the metallographic method; however, both methods confirm that the delta ferrite content decreases with a decrease in the linear energy supplied to the joint. The lowest delta ferrite content was obtained in the EBW joint—0.7%, and the highest for the TIG joint—4.9%. The properties of welded joints made of austenitic steels depend largely on the content of delta ferrite in the welds. In the case of industrial structures, the presence of ferrite in appropriate amounts is desirable. On the one hand, it increases the resistance of the weld to hot cracking and promotes refinement of the structure in the weld, but on the other hand, it reduces plasticity and impact strength at negative temperatures. The amount of ferrite in austenitic welds must, therefore, be strictly controlled. It is assumed that in order to ensure optimal technological strength of the joints, the amount of delta ferrite should be within the range of 2–10% [36,37,38,39]. Uneven amounts of energy introduced into the joints using different welding methods were clearly revealed when measuring the delta ferrite content along the length and in various areas of the cross-section of the sample (Figure 9).

The results of the X-ray phase composition analysis for the base material and joints made of AISI 316Ti steel are presented in Figure 10. Apart from the peaks originating from the austenite matrix and delta ferrite, no other phases were identified. These results are consistent with the results obtained in this work [29]. It should be mentioned that the intensity of the peaks originating from delta ferrite increases with the increase in the linear energy supplied to the joints.

Figure 11A and Table 6 show the results of the tensile test of all analyzed welded joints. The joint welded with a consumable electrode in a protective gas shield (MIG) achieved a tensile strength of 616 MPa, a yield strength of 335 MPa and an elongation of 52%. The tension curves were similar throughout the entire deformation range. Lower strength and plastic properties were observed in works in which the authors joined 316L austenitic steel using the MIG method [1,40]. The tensile curves of the joint samples obtained by non-consumable electrode welding in a protective gas shield (TIG) are presented in Figure 11B. The tensile strength for this method was 605 MPa, compared to the joint obtained using the MIG 316Ti method, this value was lower by approximately 11 MPa. The yield strength was 333 MPa and the elongation was 46%. Compared to the mechanical properties of TIG joints obtained in [16,41], the tensile strength in this work was higher, approximately 50 MPa. Each sample fractured in the base material outside the weld area. Figure 11C shows the tensile curves of samples taken from a joint obtained by plasma welding (PAW). Compared to other joints made of AISI 316Ti austenitic steel presented in this work, the strength properties obtained from the tensile test were the lowest. The average tensile strength in this case was 574 MPa and the yield strength was 315 MPa. However, no major differences were found in the elongation, the average value of which was 42%. The tensile curves of samples taken from an electron beam welded joint made of AISI 316Ti steel are presented in Figure 11D. In the Electron Beam Welding method, the average tensile strength of the samples reached 605 MPa, while the yield strength and tensile strength were 325 MPa and 42%, respectively. The authors reported similar results in [31,42], where they joined sheets of 316L steel using Electron Beam Welding.

The hardness results obtained on all cross-sections are shown in Figure 12. In the case of the MIG joint (Figure 12A), it was found that the highest hardness (approx. HV 190) was recorded in the weld axis. The average hardness in the remaining areas of the joint was HV 179 for the heat-affected zone and HV 160 for the base material, respectively. The increase in hardness both in the HAZ zone and in the area of the weld axis is related to the increased content of delta ferrite, which was confirmed by microstructural tests and measurement of the delta ferrite content (Figure 9). Figure 12B shows the hardness distribution map of the TIG 316Ti joint. Similar to the hardness results for the MIG 316Ti joint, the highest hardness was recorded in the central part of the weld. The maximum hardness in this area was HV 200 and this value was 25% higher than the hardness of the base material. From the obtained 3D hardness map, it can be seen that the hardness in the joint area is not uniform; is higher in areas where the cooling rate was higher compared to other areas of the joint, which resulted in a higher delta ferrite content. The hardness results of the welded joint obtained by plasma welding are shown in Figure 12C. Compared to the hardness results obtained for MIG and TIG joints, the hardness difference between the base material and the center of the weld axis was the lowest and amounted to 6 HV. A similar dependence on the increase in hardness in the joint area was observed in [23,24,28]. Figure 12D shows the hardness distribution map for the EBW 316Ti welded joint. The average hardness of the base material was HV 160. Towards the weld axis, the hardness increased in the heat-affected zone, where the hardness was HV 184. The maximum hardness was found in the central part of the weld, the average hardness in this area was higher by HV 28 compared to the starting material and amounted to 188 HV. The hardness map reflects the increased hardness in the entire joint area compared to the base material. The most intense red color is located in the area of the weld face, which means that the hardness is the highest in this area. The hardness tests conducted in the joints showed that higher hardness was found in the weld area and the HAZ zone. Similar hardness results have been observed in numerous scientific and research works [23,24,31,42].

## 4. Conclusions

The presented test results of various welding methods of 316Ti steel did not reveal any defects or welding imperfections in all the analyzed welded joints.Mechanical properties tests showed that the produced joints achieve a similar level of tensile strength compared to the base material. The joint efficiency was 93% for the plasma welding method and 98% in the other welding methods.In all analyzed joints, the amount of delta ferrite in the face area was higher compared to the weld root.The least amount of delta ferrite was found in the 316Ti steel weld made using the EBW method—approximately 2%; in the MIG method, it was approximately 4%, and the remaining TIG and PAW welds contained 4–5% of delta ferrite.It has been shown that the average hardness of joints in 316Ti (160 HV) steel is approximately 30 units higher at the point of highest hardness compared to the base material (210 HV).

## Figures and Tables

**Figure 1 materials-17-01681-f001:**
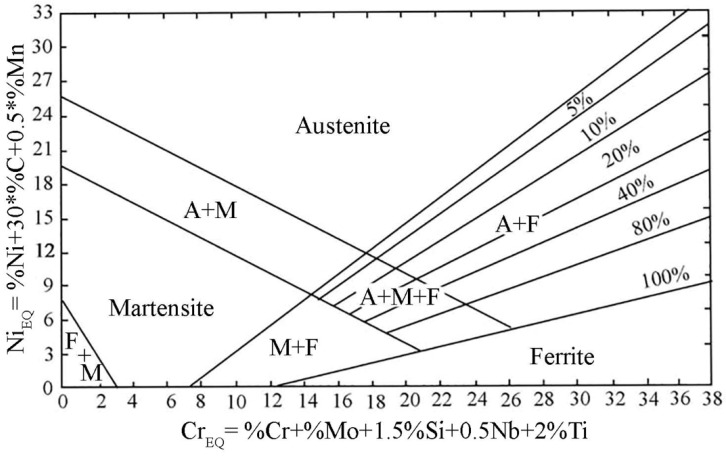
Schaeffler graph showing the share of austenite and ferrite in corrosion-resistant steels [11].

**Figure 2 materials-17-01681-f002:**
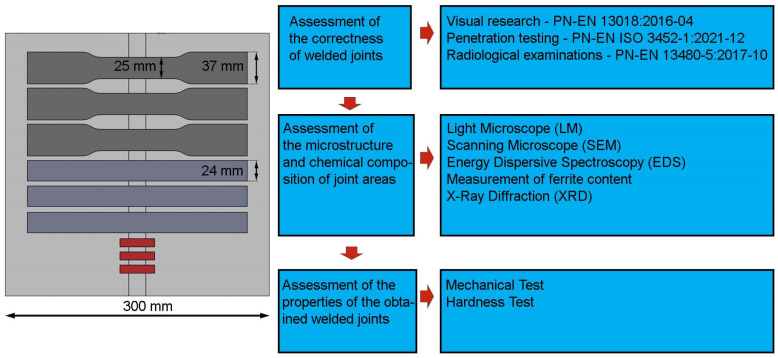
Test scheme and dimensions of test samples.

**Figure 3 materials-17-01681-f003:**
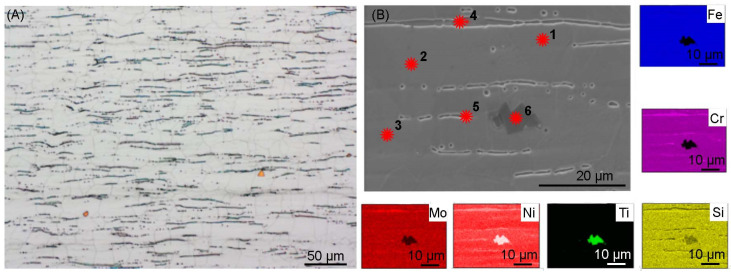
(**A**) Microstructure of AISI 316Ti steel, light microscopy; (**B**) microstructure of AISI 316Ti steel (SEM) with marked places of EDS point analysis of the chemical composition, elements distribution map.

**Figure 4 materials-17-01681-f004:**
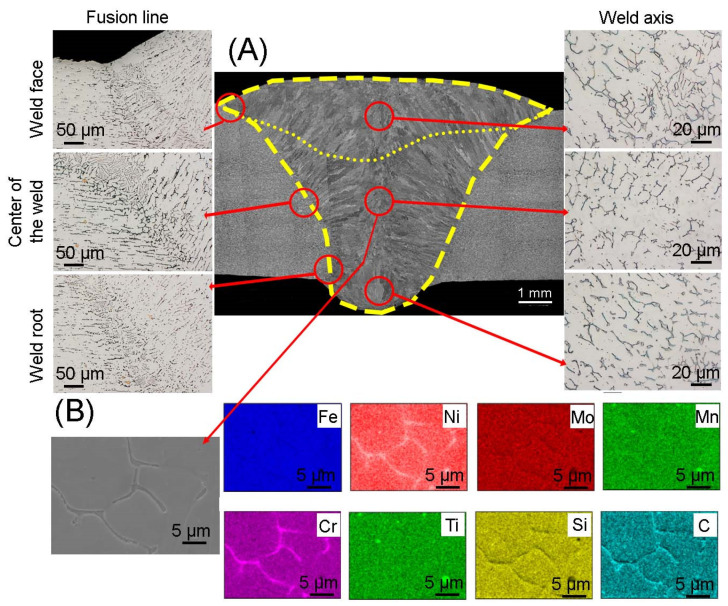
(**A**) Microstructure of the joint obtained using the MIG method; (**B**) elements distribution map in the area of the weld center.

**Figure 5 materials-17-01681-f005:**
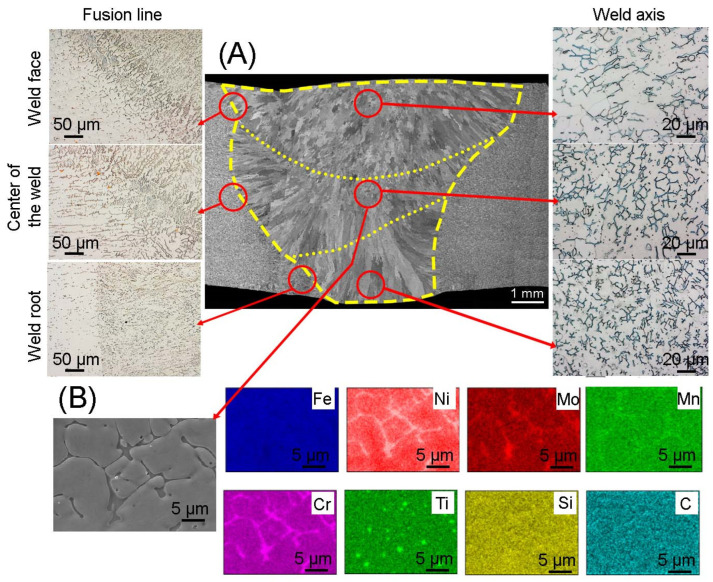
(**A**) Microstructure of the joint obtained using the TIG method; (**B**) elements distribution map in the area of the weld center.

**Figure 6 materials-17-01681-f006:**
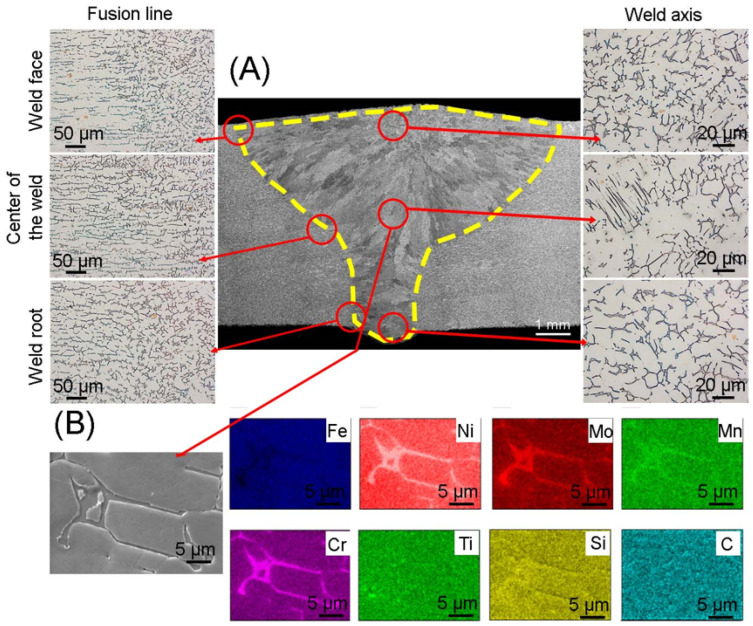
(**A**) Microstructure of the joint obtained using the PAW method; (**B**) elements distribution map in the area of the weld center.

**Figure 7 materials-17-01681-f007:**
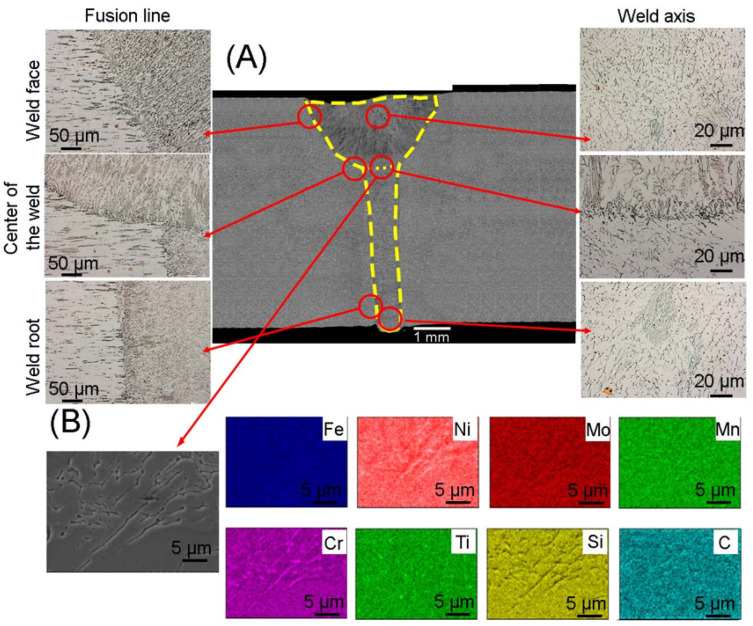
(**A**) Microstructure of the joint obtained using the EBW method; (**B**) elements distribution map in the area of the weld center.

**Figure 8 materials-17-01681-f008:**
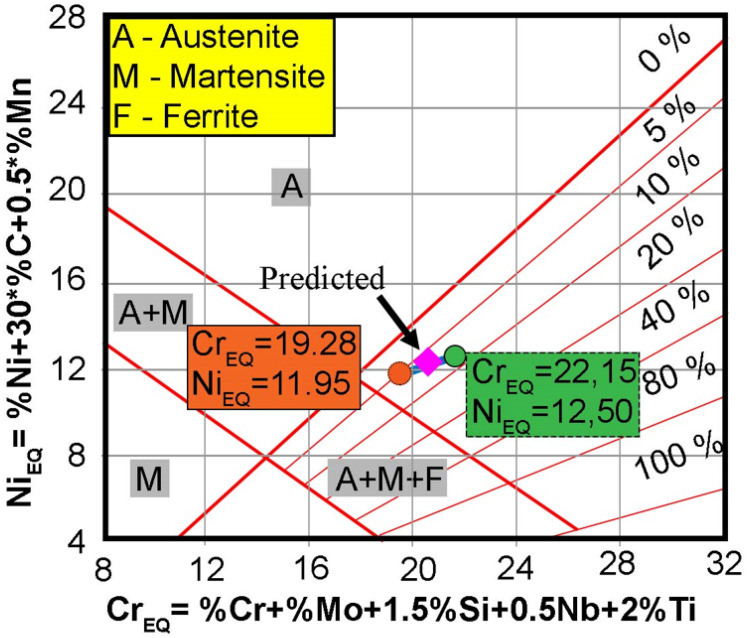
Base metals and fusion zone represented on the Schaeffler diagram.

**Figure 9 materials-17-01681-f009:**
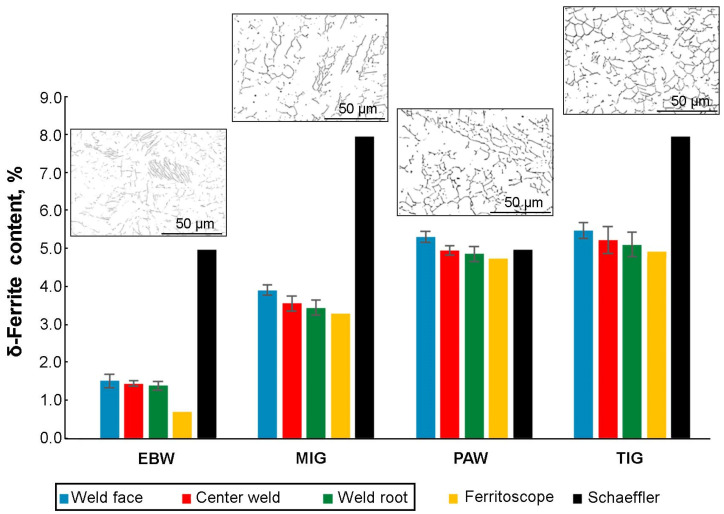
The amount of delta ferrite in the obtained joints determined metallographically, using a ferritometer and a Schaeffler diagram.

**Figure 10 materials-17-01681-f010:**
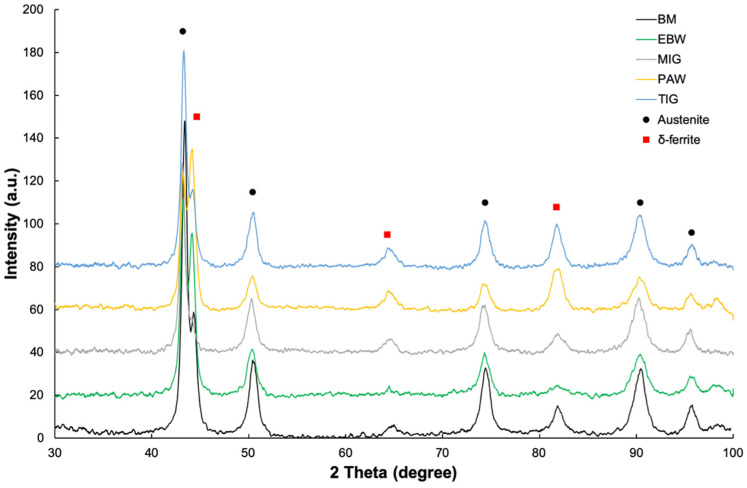
Results of X-ray phase analysis (XRD) of the obtained joints.

**Figure 11 materials-17-01681-f011:**
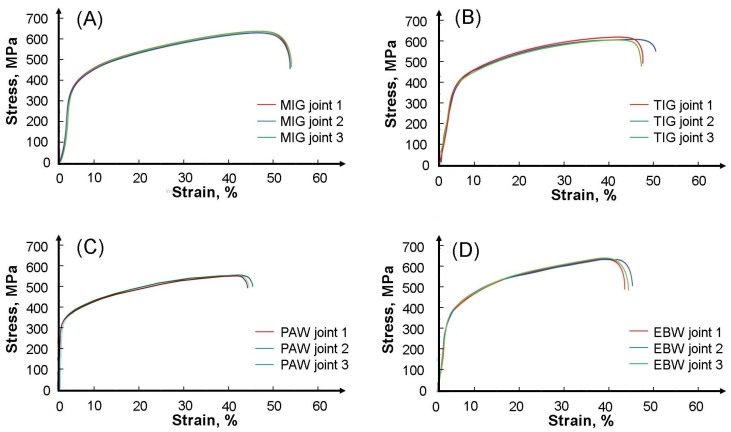
Tensile curves of joints: (**A**) MIG, (**B**) TIG, (**C**) PAW, and (**D**) EBW.

**Figure 12 materials-17-01681-f012:**
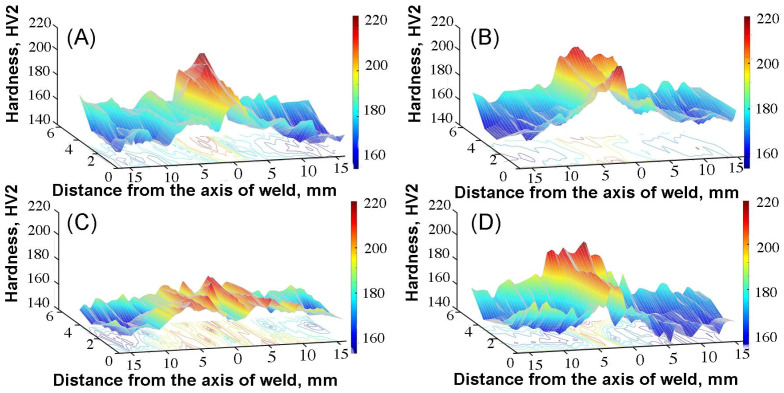
Hardness maps for joints: (**A**) MIG, (**B**) TIG, (**C**) PAW, and (**D**) EBW.

**Table 1 materials-17-01681-t001:** Phases formed when austenitic steels are held at high temperatures.

Phase	Type	Composition	Formation Temperature Range
Chromium carbides	M_23_C_6_	(Cr, Fe, Mo)_23_C_6_	600–950 °C
Chromium carbides	M_6_C	(Cr, Fe, Mo)_6_C	700–950 °C
Chromium nitrides	MN	(NbCr)N	700–1000 °C
Sigma	AB	(Fe, Cr, Mo, Ni)	550–1050 °C
Chi	A_48_B_10_	Fe_36_Cr_12_Mo_10_ (FeNi)_36_Cr_18_(TiMo)_4_	600–900 °C
Laves	A_2_B	(FeCr)_2_(Mo, Nb, Ti, Si)	550–900 °C

**Table 2 materials-17-01681-t002:** Chemical composition of austenitic steel 316Ti, weight%.

Material	C	Si	Mn	P	S	Cr	Ni	Mo	Ti	N	Fe
**AISI 316Ti**	0.033	0.42	0.92	0.038	0.001	16.6	10.5	2.02	0.3	0.013	69.155

**Table 3 materials-17-01681-t003:** Chemical composition of AISI 316Si welding wire, weight%.

Material	C	Si	Mn	P	S	Cr	Ni	Mo	Cu	Fe
**AISI 316Si**	0.01	0.83	1.60	0.02	0.001	18.3	11.4	2.60	0.2	65.039

**Table 4 materials-17-01681-t004:** Welding parameters.

Run	Diameter of the Filler Material Φ [mm]	Welding Current I [A]	Arc Voltage U [V]	Welding Speed Vs [mm/min]	Linear Welding Energy Q [kJ/mm]
**MIG 316Ti**					
1	1.2	150	18.5	350	0.28
2	1.2	175	21	350	0.39
**TIG 316Ti joint**					
1	2.0	170	12	150	0.65
2	2.0	170	12	150	0.69
3	2.0	170	12	150	0.69
**PAW 316Ti**					
1	-	190	25	160	1.49
**EBW 316Ti joint**					
1	-	0.0195	120	800	0.14
2	-	0.010	120	500	0.11

**Table 5 materials-17-01681-t005:** Results of the analysis of the chemical composition of EDS in the places marked in Figure 3B.

Base Material AISI 316Ti	Si	Ti	Cr	Mn	Fe	Ni	Mo
**Point 1**	0.56	0.17	17.05	1.12	69.31	9.83	1.97
**Point 2**	0.55	0.16	16.45	1.16	68.87	11.05	1.76
**Point 3**	0.62	0.17	16.85	0.95	68.92	10.58	1.86
**Point 4**	0.64	0.17	22.90	0.01	67.55	5.11	3.62
**Point 5**	0.54	0.66	23.30	0.92	67.07	4.30	3.21
**Point 6**	0.11	97.61	0.69	0.14	1.30	0.16	0.00

**Table 6 materials-17-01681-t006:** The results of mechanical properties determined on the basis of Figure 11.

	R_m_ [MPa]	R_p0.2_ [MPa]	A [%]
**MIG 316Ti**	616	334	52
**TIG 316Ti**	605	332	46
**PAW 316Ti**	574	315	42
**EBW 316Ti**	605	325	42

## Data Availability

Data are contained within the article.

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
