# Peer review of "The Influence of Various Welding Methods on the Microstructure and Mechanical Properties of 316Ti Steel"

_materials, 2024, doi:10.3390/ma17071681_

Round 1

Reviewer 1 Report

Comments and Suggestions for Authors

The presented manuscript seems to be interesting for readers of the Materials journal, it is written in a good manner and suits the requirements of the journal. It can be accepted for publication after the minor corrections listed below. 

- The "Abstract" section should contain the main achievements of research, not a general discussion. Re-organization of the abstract is needed.

- The dictation of the words martensite, ferrite, and austenite in Figure 1 should be corrected

- Figure 7 should be referenced within the text. and give more explanations about it

- The vertical axis of Figure 7 should be changed to Ni(eq) instead of Cr(eq).

- The Cr(eq) relationship on the horizontal axis of Figure 1 and Figure 7 should be the same.

- 1 out of 37 references are for after 2020, while 5 references are for before 2000.

- “Imperfections” should be changed to “imperfections”. Also “Lavesa“ should be changed to “Laves“

- The novelty of the work at the end of the manuscript “introduction” is not sufficient and should be explained more.

- The reason for choosing the welding parameters in each of the methods should be presented or referenced

- The absence of defects in any of the methods must be proven with sufficient evidence

- The flowchart of the research method should be given. Also, sample coding and specifications should be provided in the table.

- All parameters used in formulas must be explained. It is recommended to attach all parameters and abbreviations used in a table at the end of the article.

- Abbreviations/ acronyms, should all be defined at their first occurrence in the manuscript; for example TIG, MIG, PAW

- In the "Conclusion" section, the authors should present more quantitative data as the main results of the research study rather than just some qualitative data. 

- Literature review is not sufficient and authors must review and cite more papers in the field. Doing this, reviewing the following refs could be helpful for the given discussions: doi.org/10.1590/S1516-14392013005000170, doi.org/10.1007/s11665-017-3095-7

Author Response

Dear Reviewer,

The responses to the comments can be found in the attachment

Reviewer 2 Report

Comments and Suggestions for Authors

The authors have chosen a relevant topic for their research and the findings of the paper are of importance for the scientific community especially the industrial field. The research is well presented and composed. However, the following are the review comments, which the authors shall address to increase the overall quality of their paper. These comments are inclusive of the grammatical and technical comments.

•             The authors are requested to thoroughly check their paper to match the format, font, style, line spacing, instructions, references, etc. as given in the standard paper template. Example: Use of colour for headings.

·                  Abstract should be summarized in a better way.

·                  Introduction is too long. Please summarize it better.

•             It is suggested to take care of figure captions as per template.

•             It is suggested to increase the clarity of figure 1 to improve readability.

The section materials and methods is not clear. More details are needed.

•             It is suggested to present the conclusions point wise highlighting the findings of the study.

•             The authors are requested to thoroughly check their paper for grammatical errors and technical language.

Author Response

(The authors gave the same response as above.)

Reviewer 3 Report

Comments and Suggestions for Authors

1.What do the different colors on Figure 10 represent? They need to be labeled in the figure.

2.Part of the line segments in Figure 10 overlap and can be zoomed in locally.

3.The author can use more data lists to reflect the experimental results, otherwise the author's experimental results cannot be explained from the experimental data.

4. The author should provide an explanation of the experimental conditions.

5.There are many grammar errors. Please check English carefully through the paper.

Comments on the Quality of English Language

The language quality of the article can be understood professionally, and slight modifications are needed to make its expression more fluent.

Author Response

(The authors gave the same response as above.)

Round 2

Reviewer 1 Report

Comments and Suggestions for Authors

The revision is acceptable

Reviewer 2 Report

Comments and Suggestions for Authors

The revised version of the paper looks more better. The authors have addressed all comments successfully. The paper could be accepted for publication in it’s current form.